# Assessing the potential of a Bayesian ranking as an alternative to consensus meetings for decision making in research funding: A case study of Marie Skłodowska-Curie actions

Rachel Heyard[1]*, David G. Pina[2], Ivan Buljan[3], Ana Marušić[4]

1 Center for Reproducible Science, Epidemiology, Biostatistic and Prevention Institute, University of Zurich, Zurich, Switzerland, 2 European Research Executive Agency, European Commission, Brussels, Belgium, 3 Department of Psychology, Faculty of Humanities and Social Sciences, University of Split, Split, Croatia, 4 Center for Evidence-based Medicine, Department of Research in Biomedicine and Health, University of Split School of Medicine, Split, Croatia

* rachel.heyard@uzh.ch

**Data availability statement:** The statistical analyses were conducted using the R programming language, version 4.4.0. The Bayesian ranking methodology was implemented using the R package

## Abstract

Funding agencies rely on panel or consensus meetings to summarise individual evaluations of grant proposals into a final ranking. However, previous research has shown inconsistency in decisions and inefficiency of consensus meetings. Using data from the Marie Skłodowska-Curie Actions, we aimed at investigating the differences between an algorithmic approach to summarise the information from grant proposal individual evaluations to decisions after consensus meetings, and we present an exploratory comparative analysis. The algorithmic approach employed was a Bayesian hierarchical model resulting in a Bayesian ranking of the proposals using the individual evaluation reports cast prior to the consensus meeting. Parameters from the Bayesian hierarchical model and the subsequent ranking were compared to the scores, ranking and decisions established in the consensus meeting reports. The results from the evaluation of 1,006 proposals submitted to three panels (Life Science, Mathematics, Social Sciences and Humanities) in two call years (2015 and 2019) were investigated in detail. Overall, we found large discrepancies between the consensus reports and the scores a Bayesian hierarchical model would have predicted. The discrepancies were less pronounced when the scores were aggregated into funding rankings or decisions. The best agreement between the final funding ranking can be observed in the case of funding schemes with very low success rates. While we set out to understand if algorithmic approaches, with the aim of summarising individual evaluation scores, could replace consensus meetings, we concluded that currently individual scores assigned prior to the consensus meetings are not useful to predict the final funding outcomes of the proposals. Following our results, we would suggest to use individual evaluations for a triage and subsequently not discuss the

ERforResearch, which is publicly accessible at https://github.com/snsf-data/ERForResearch. The Bayesian hierarchical models for the Bayesian ranking were fitted using JAGS, which requires prior installation. The code and data required to reproduce the analyses are openly available at https://gitlab.com/heyardr/msca-br-comparison. Additionally, a snapshot of the repository at the time of writing is preserved and accessible via Zenodo at https://doi.org/10.5281/zenodo.11192507.

**Funding:** Ana Marušić receives funding from the Croatian Science Foundation under Grant agreement No. IP-2019-04-4882. The funder had no role in the design of this study, its execution, analyses, interpretation of the data, or decision to submit results. The European Research Executive Agency (REA) has provided financial support for the APC fees.

**Competing interests:** I have read the journal's policy and the authors of this article have the following competing interests: Rachel Heyard conceptualized and implemented the Bayesian ranking methodology used in this study while employed at the Swiss National Science Foundation. David G. Pina is employed by the European Research Executive Agency. Ivan Buljan declares no competing interests. Ana Marušić occasionally serves as an expert evaluator of grant proposals for the European Commission. This does not affect our adherence to PLOS ONE policies on data and materials sharing.

weakest proposals in panel or consensus meetings. This would allow a more nuanced evaluation of a smaller set of proposals and help minimise the uncertainty and biases when allocating funding.

## Introduction

Funding agencies strive to fund the most excellent research projects while considering their limited funding budget. To this end, they rely on expert evaluations to make decisions on which proposals to approve and which to reject. However, grant peer review has been criticised for being inefficient and biased [1–3]. Panel meetings and discussions on grant and researcher evaluation have been referred to as a "black-box", because the dynamics leading to the funding decisions are not fully understood [4]. Further, important discrepancies may exist between individual scores given prior to a panel meeting and the final scores on which the funding decisions are based [5]. Such discrepancies might even conceal unequal evaluation of certain demographics, *e.g.*, gender differences in individual scores, because panels rectify such imbalances in their funding decisions [6]. On top of that, onsite panel meetings or consensus meetings have been deemed inefficient, too expensive and time consuming [7]. The aforementioned drawbacks, coupled with the inherent limitations of grant peer review, have prompted discussions to employ alternative approaches to funding allocation, including the integration of lottery elements in the processes [8,9].

The procedures used by funding agencies to allocate funding are often simplistic and ignore the underlying uncertainty [10]. Specifically, many funders aggregate evaluations using simple averages, quintiles or medians of the scores given by experts as a basis for proposal ranking and funding decisions (see for example the process of the National Institutes of Health discusses in [11]). However, given the documented evidence of low inter-rater reliability in the evaluation of grant proposals [12,13], relying on simple summary statistics, while ignoring the uncertainty and variability that is present in the scores, is ill-advised. Other funders use panel meetings or consensus meetings to summarise individual expert evaluations. The European Commission, for example, uses such consensus meetings to establish consensus scores for the evaluation of research proposals submitted under its Framework Programme for Research and Innovation [14]. In general, absolute ranking schemes based on simple summaries of evaluation scores can lead to situations where very small differences in the average scores or consensus scores, likely due to uncertainty or natural variation and not to actual quality differences, result in one proposal receiving funding while the next, equally good, is being denied funding. Including lottery elements in the process could resolve part of the concern by explicitly acknowledging the role of chance and uncertainty [15–17]. Lottery elements could also ease the workload for experts, eliminating the necessity for additional review cycles or discussions of the proposals in the "grey area" of equally good proposals [18]. The Swiss National Science Foundation (SNSF) recently introduced a comparative ranking strategy called the Bayesian ranking [BR,19]. The strategy uses a Bayesian hierarchical model which compares each proposal against every other proposal while accounting for the uncertainty inherent to the evaluation process and directly augments the funding allocation process with a lottery component. However, generally, the dynamics of panel or consensus meetings are not understood well enough to determing whether replacing the meetings with such model based aggregation and ranking methods are warranted.

Therefore, the present study uses, as a sample, the Marie Skłodowska Curie Actions (MSCA) Innovative Training Networks (ITN) which funds the mobility and training of

young researchers in Europe. We aim to investigate the extent to which the MSCA consensus meetings could be substituted by an algorithmic aggregation of individual evaluation scores assigned prior to the consensus meeting. The MSCA consensus meetings aim at establishing a funding ranking, grouping the proposals into the main list of funded proposals, the reserve list of proposals to be funded if additional funding becomes available, and the rejected list. We specifically use the BR to determine the contexts in which such a substitution would be advantageous and the scenarios in which it may be less suitable. This study provides valuable insights into whether the outcomes or decisions derived from the consensus meeting can be predicted from the individual evaluation scores alone, or whether the dynamics of the consensus meetings are unpredictable solely from the individual evaluation scores. Our results contribute to enhancing our knowledge of the feasibility and effectiveness of such algorithmic approaches in grant proposal evaluations, ultimately facilitating evidence-based decision-making in research funding allocation and making the process more transparent and efficient for expert evaluators, panel members and applicants.

## Research questions

Our analysis aimed at addressing the following three research questions:

(1) Can a Bayesian hierarchical model predict the consensus report using the individual evaluation reports given by the experts?
(2) Can the Bayesian ranking mitigate the consensus meetings?
(3) How do the proposals on the final MSCA reserve list differ from the proposals in the BR lottery group?

## Methods

### Data source: The MSCA European Training Networks

Under Horizon 2020, the EU Framework Programme for Research and Innovation that ran from 2014 to 2020, the MSCA ITN programme supported joint research training or doctoral programs implemented by partnerships across Europe, and beyond. These partnerships were taking the form of collaborative European Training Networks (ETN), European Industrial Doctorates (EID) or European Joint Doctorates (EJD). Note that after 2020, the next EU Framework Programme started, Horizon Europe, and since the comparability of the programmes is unclear, this study focuses on the fully completed Horizon 2020 programme. Additionally, only the ETN, the largest of these 3 funding schemes, is included. The MSCA evaluation process and subsequent funding allocation is represented in the top layer of Fig 1. Each proposal was individually evaluated and scored by at least three experts on three different criteria: scientific excellence (STE), impact (IPT) and implementation (IPL). While all proposals were scored by three experts, in very rare cases, when the experts were not able to agree on a consensual evaluation, a forth expert could be consulted. A continuous scale, from 0 to 5 where 0 is the lowest and 5 is the best score, was used for each criterion. The criteria scores were weighted into individual total scores using the formula 50%STE + 30%IPT + 20%IPL and converted to a scale from 0 to 100 (i.e., individual total scores $\times 20$). These converted scores represent the individual evaluation reports (IER) in the far left of Fig 1. After the individual scoring, the experts met and discussed the proposals to reach a consensus score for each criterion, which were again converted into a consensus total score through the

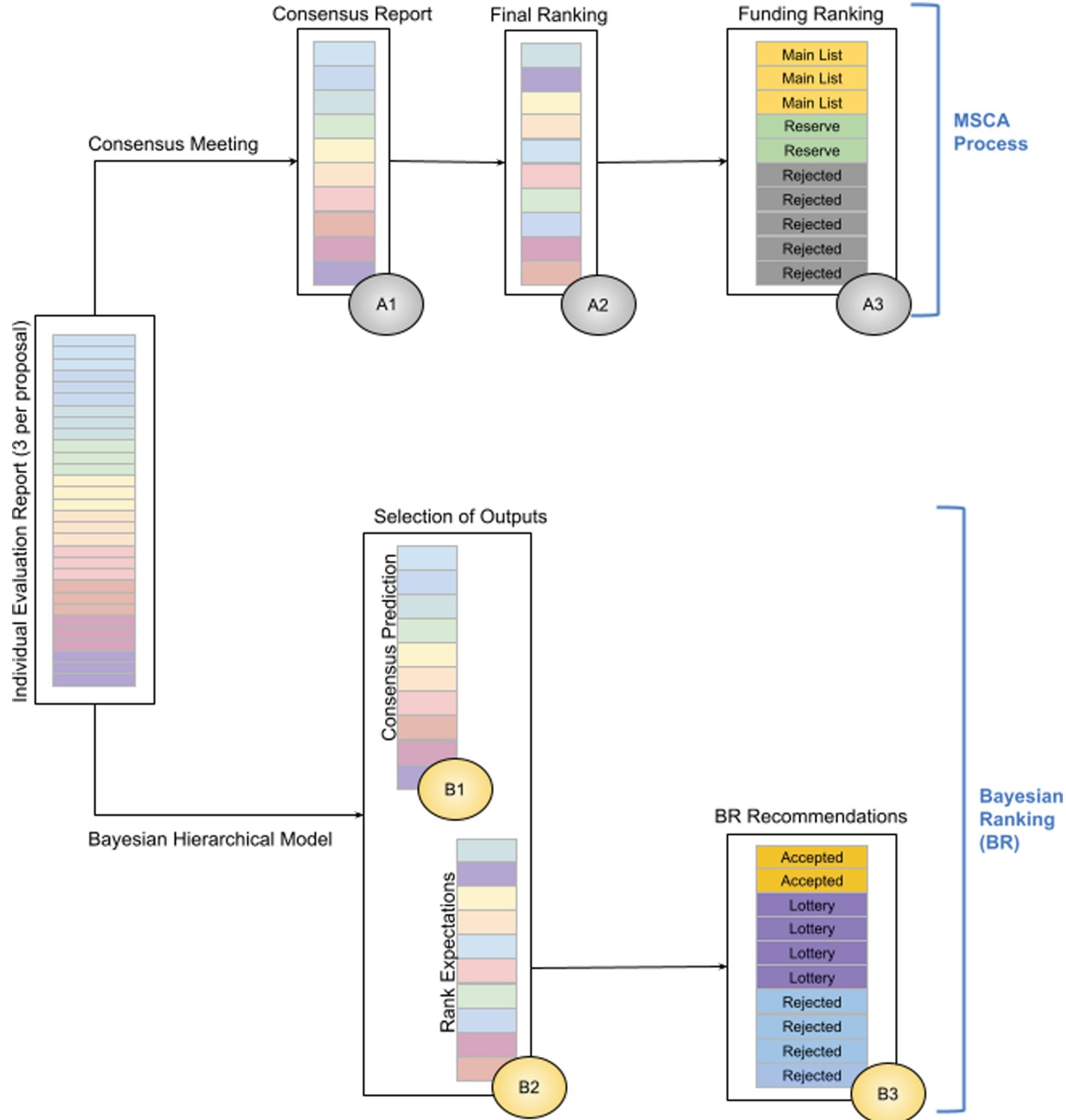

**Fig 1. In the top layer of the Figure, the MSCA funding allocation process is represented for ten hypothetical proposals: the individual evaluation reports established by three experts to each proposal are followed by a consensus report prepared during the consensus meeting (A1).** The consensus total scores are then used to compound the final ranking (A2). The final ranking, established per scientific panel, and considering the available budget, determines the proposals that will be included in the three funding ranking groups: the main list, the reserve list and the rejected proposals (A3). Similarly, the bottom layer represents the Bayesian ranking process for the same hypothetical proposals: the individual total scores from the experts' individual evaluation reports are used in the Bayesian hierarchical model intended to model the evaluation process. From this model, various outputs can be extracted, two of which are predictions of the consensus reports established for each of the ten proposals (B1) and the expectations and distribution of the final rankings (B2). Combining the BR output with the available budget gives us the BR recommendations, allocating the proposals to either the accepted, lottery or rejected group (B3).

same formula and summarised in a consensus report (CR, A1 in Fig 1). After the consensus meeting each proposal achieved a unique consensus score for the three criteria. The proposal final ranking, A2 in Fig 1, was based on the consensus total score derived from the consensus report. For the MSCA funding ranking (A3 in Fig 1), the proposals were either ranked on the main list of accepted proposals, the reserve list of proposals that could be funded in case some additional funding becomes available, or on the rejected list. The MSCA funding ranking was based on the final ranking considering the available funding budget of the panel.

In total, 7,870 proposals were evaluated in the context of MSCA ETN between the calls in 2014 and 2019 within eight scientific panels (Chemistry, Economics, Engineering, Environment and Geo Sciences, Life Sciences, Mathematics, Physics and Social Sciences/Humanities). For feasibility of this exploratory study, we selected two calls and three panels. More specifically, the proposals submitted for the calls in 2015 and 2019, and to the panels Mathematics, Social Sciences/Humanities and Life Sciences were retained for our analysis. As such, the analysis included one small panel (Mathematics evaluating 0 – 5% of all proposals), one medium panel (Social Sciences/Humanities evaluating 6 – 20% of all proposals) and one large panel (Life Sciences evaluating >20% of all proposals) from an earlier and a more recent call. Table 1 summarises the data from the evaluation of a total of 1,006 in the three scientific panels and the two calls.

The Life Sciences panels were the largest, with the largest budget, and even though the panels were composed of more experts, each expert also had to evaluate on average a higher number of proposals compared to experts in smaller panels. Importantly, Table 1 further shows the share (in percentage) of proposals distributed into the different MSCA funding lists: the main list of proposals retained for funding, the reserve list and the list of rejected proposals. Since the budget available per panel is chosen to be proportional to the number of proposals submitted to the different panels, these shares are comparable across panels. The data also includes information on the country of the proposal's coordinating organisation and the number of participating organisations involved in the proposal, which can be found in the raw data.

## Methods for proposal ranking

Different methods can be used to rank proposals. We compared the official MSCA ranking procedure relying on consensus meetings to the results of the Bayesian ranking procedure. More specifically, our analysis compared the following two ranking methods:

1. First, each proposal received an official final and unique rank given by the respective MSCA panel (as in A2 in Fig 1), which is established using the consensus reports (A1) and is the basis for the final funding ranking (A3). Note that, to ensure the uniqueness of the ranking, i.e., to avoid ties, if a set of proposals had the same consensus total score, the consensus score given to the excellence criterion prevailed. Whenever this condition was not sufficient to uniquely rank the proposals, the consensus score given to the impact criterion was used. The final ranking and funding ranking were retrieved from the raw data.

2. A second ranking of proposals was established using the Bayesian ranking (BR) procedure described in [19]. For this, a Bayesian hierarchical model was implemented which relates the individual evaluation reports, $y_{ij}$ given by expert $j$ to proposal i, to the true quality of the proposal $\theta_i$ and the expert effect $\lambda_j$. In this setting, the individual evaluation report $y_{ij}$ can be interpreted as the expert's best guess or estimation of

**Table 1. Number of proposals evaluated by the selected panels (Mathematics, Social Science/Humanities and Life Sciences) for the two selected calls (2015 and 2019), together with the percentage of successful and unsuccessful proposals, and proposals on the reserve list.** The number of experts per panel and the average number of proposals evaluated by each expert with its minimum and maximum number are represented. The median budget requested by the proposals in the panels for the two calls, together with the budget that is available and can be distributed is also shown. Soc. Sci./Hum.: Social Sciences and Humanities, Av: Average, N: Number.

| | N | Main list (%) | Reserve list (%) | Rejected (%) | N experts | Av n of proposals per expert [min–max] | Median requested budget (million) | Available budget (million) |
|---|---|---|---|---|---|---|---|---|
| **Call year: 2015** | | | | | | | | |
| Mathematics | 18 | 1 (5.6) | 2 (11.1) | 15 (83.3) | 11 | 4.9 [1–11] | 3.8 | 4.3 |
| Soc. Sci./Hum. | 114 | 7 (6.1) | 3 (2.6) | 104 (91.2) | 52 | 6.6 [1–11] | 3.7 | 27.4 |
| Life Sciences | 381 | 24 (6.3) | 10 (2.6) | 347 (91.1) | 129 | 8.9 [1–15] | 3.6 | 91.6 |
| **Call year: 2019** | | | | | | | | |
| Mathematics | 16 | 1 (6.2) | 1 (6.2) | 14 (87.5) | 14 | 3.4 [1–10] | 3.9 | 5.1 |
| Soc. Sci./Hum. | 122 | 10 (8.2) | 3 (2.5) | 109 (89.3) | 76 | 4.8 [1–10] | 3.8 | 38.9 |
| Life Sciences | 355 | 27 (7.6) | 12 (3.4) | 316 (89) | 187 | 5.7 [1–11] | 3.9 | 113.3 |

the proposal's true quality $\theta_i$. The expert effect $\lambda_j$ describes the deviation from the overall mean $\bar{y}$ that can be attributed to the specific expert evaluating the proposal. This deviation may be influenced by the expertise, experience and/or (implicit) biases of the expert. The remaining deviation from the overall mean is captured in the parameter for the true quality $\theta_i$: a positive $\theta_i$ suggests that the proposal $i$ is of better quality than the average proposal evaluated in the same call. The underlying Bayesian hierarchical model is defined as follows

$$
\begin{aligned}
y_{ij} \mid \theta_i, \lambda_j &\sim N(\bar{y} + \theta_i + \lambda_j, \sigma^2) \\
\theta_i &\sim N(0, \tau_\theta^2) \\
\lambda_j &\sim N(0, \tau_\lambda^2).
\end{aligned}
\tag{1}
$$

The prior distribution of $\theta_i$ is centred around 0 assuming that the proposals enter the evaluation with the same a priori chance of achieving a high or a low score. In [19], the expert effect represented by $\lambda_j$ depended on the proposal, and its prior was not necessarily centred around 0. This enabled the modeling of expert behaviour and a better understanding of whether a certain expert score was in-line with their general behaviour or not. This additional model complexity is not recommended in the context of the MSCA evaluation, as only limited data are available on each expert's call-specific scoring habits. Furthermore, the distribution of the parameter of interest $\theta_i$ remains unaffected by this change, thereby justifying the employment of the simplified model version with a normal prior centred around 0 for $\lambda_j$. The following uniform hyper-priors are applied on the variance parameters

$$
\begin{aligned}
\tau_\theta, \tau_\lambda &\sim U_{[0,30]}, \\
\sigma &\sim U_{[0,3]},
\end{aligned}
$$

which ensures a worst-case upper bound for the parameters as all the scores $y_{ij}$ lie in the interval $[0, 100]$. The prior on the variation of the individual scores around the linear predictor $\sigma^2$ is assumed to be less flat. Different outputs can be extracted from the Bayesian hierarchical model. The posterior means of the part of the linear predictor in Eq 1 that is independent of the expert evaluating the proposal, $\bar{y} + \theta_i$, can be used to predict the consensus report (B1 in Fig 1). Then, as outlined in [19], to rank the proposals of a certain call the expectations of the rank of the $\theta_i$'s are calculated together with their 50% credible intervals, B2 in Fig 1. A provisional funding line is defined based on the available budget for the specific call and scientific panel. Generally, a predefined budget determines the number of proposals $x$ that can be funded. A provisional funding line will be set at the expected rank (mean of the posterior distribution of the rank) of the $x$th ranked proposal. The proposals with credible interval clearly below or above the provisional funding line are directly funded or rejected. The lower the rank, the better the proposal. The proposals with credible interval crossing the provisional funding line will be subjected to a lottery group as, given the available data, those proposals' quality cannot be differentiated from the funding line. These groups form the BR recommendation as represented in B3 in Fig 1.

## Data analysis

For each research question, a specific strategy was predefined. The protocol of the planned study was preregistered on the Open Science Framework (osf.io/cnurf) before starting the data analysis. Box 1 below outlines the methodology applied to answer each research question.

Box 1. The research questions with the applied methodology to answer them.

> **Research question (1):** *Can the Bayesian hierarchical model predict the consensus report (A1 in Fig 1) using the individual evaluation reports given by the experts (B1 in Fig 1)?*
>
> The total scores from the consensus report will be visually compared to the posterior means of the part of the linear predictor in Eq (1) that is independent of the expert evaluating the proposals: $\bar{y} + \theta_i$. If the consensus reports were predictable using solely the individual evaluation reports, those two quantities should be strongly correlated. Additionally, we will compute the share of proposals with consensus reports within the 95% credible interval of what the Bayesian hierarchical model would have predicted.
>
> **Research question (2):** *Can the BR mitigate the consensus meetings? More specifically, is the Bayesian ranking similar to the true final ranking (A2 vs. B2 in Fig 1)?*
>
> The BR will be compared graphically to the final ranking based on the consensus report. The credible interval of the BR can be interpreted as the rank that would be expected. Hence, we will investigate whether the final MSCA ranking is included in the 95% BR credible interval or not. Additionally, the percentiles of the rankings will be compared by calculating the agreement between the methods, to understand whether there is more or less agreement for certain percentiles - *e.g.*, how large is the agreement between the 10% best proposals according to the BR and the best ranked 10% according to the consensus report.
>
> **Research question (3):** How do the proposals on the final MSCA reserve list differ from the proposals in the BR lottery group (A3 vs. B3 in Fig 1)?
>
> The BR recommendations "Rejected", "Lottery" and "Accepted" are compared to the MSCA funding ranking groups "Main list", "Reserve list" and "Rejected list" using contingency tables and shares of agreement.

## Results

### Data description

The distribution of the consensus scores, total and criteria, given to the proposals evaluated in the calls and panels of interest are shown in Fig 2. For all panels, scores and calls, we observe right-skewed distributions with a tendency for higher scores. Each scientific panel has its own grading behaviour which also varies from call to call, *i.e.*, the Mathematics panel seems to be more reluctant to give high scores in general, and the Social Sciences and Humanities

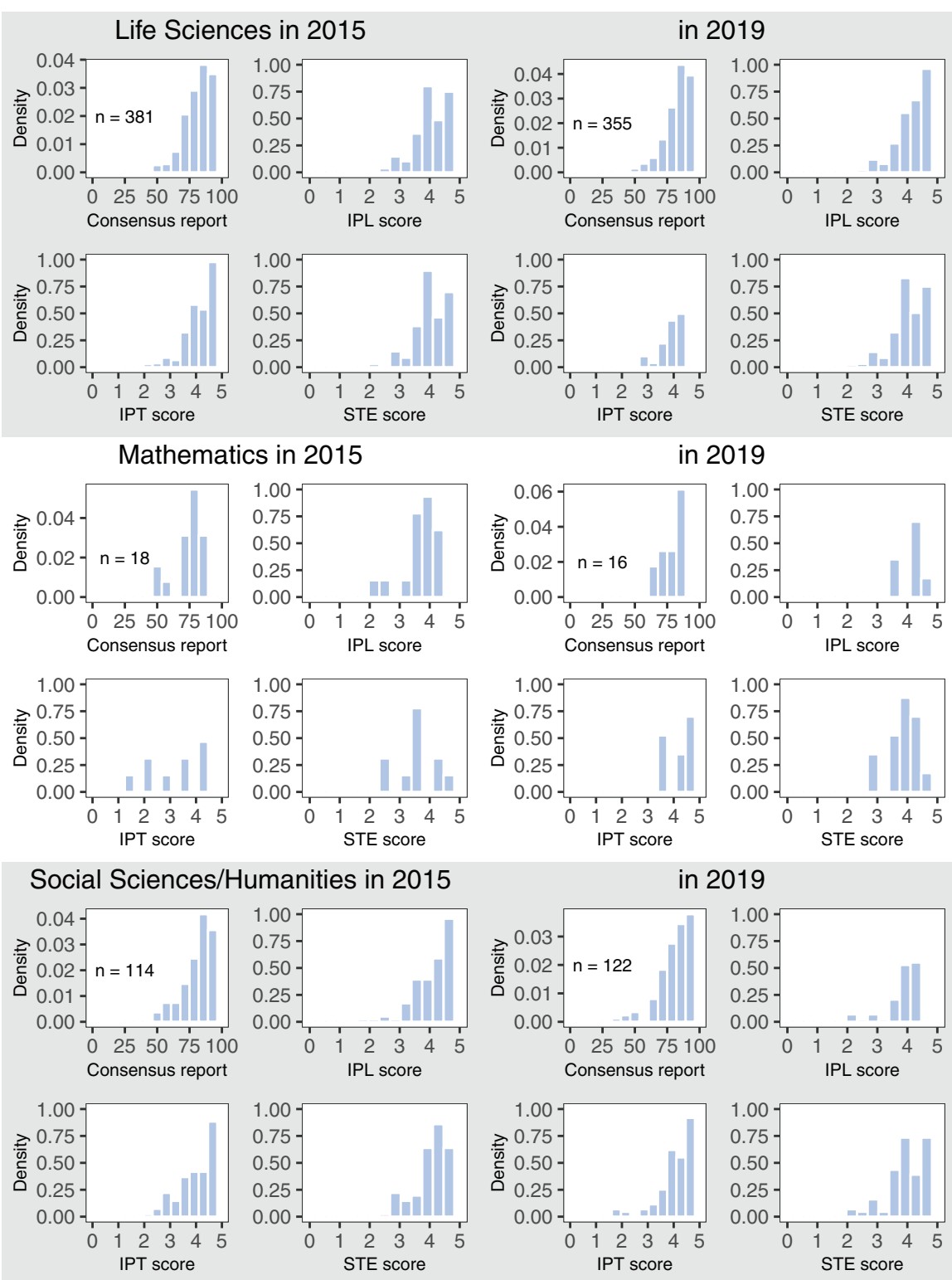

**Fig 2. Distributions of the scores after consensus meetings for the 3 different panels for the calls in 2015 and 2019.** For each panel, the first plot represents the distribution of the consensus report, while the other three plots show the distribution of the consensus scores for the different criteria. The total number of proposals evaluated in each call is also shown. STE: scientific excellence, IPT: impact, IPL: implementation.

## Social Sciences/Humanities 2015

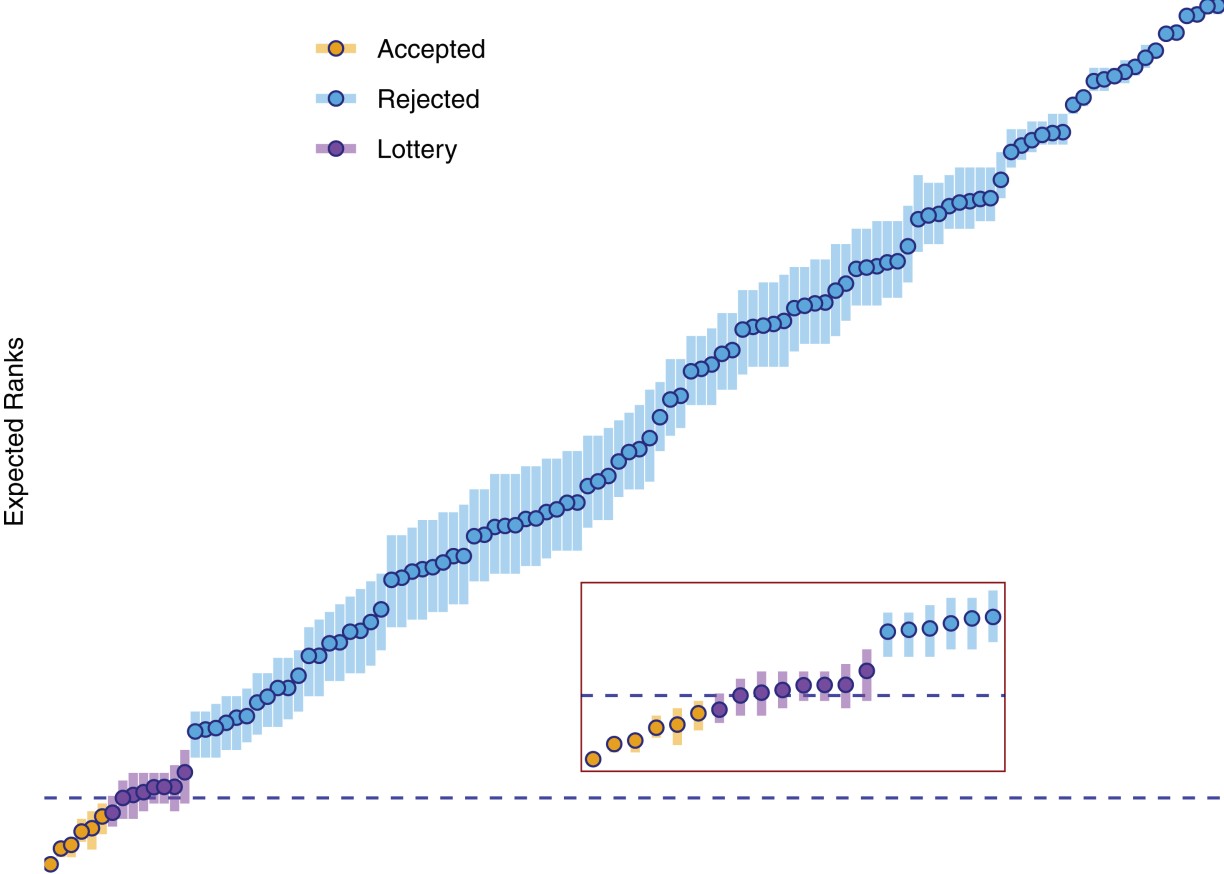

**Fig 3. Social Sciences and Humanities panel of the 2015 Call: Bayesian Ranking and recommendations.** The expected ranks are represented with their 50% credible intervals for the three panels. The provisional funding line (dashed blue line) is defined by allocating the available budget to the best ranked proposals until there is not enough funding for the next proposal. Those proposals with their 50% credible interval crossing the provisional funding line are recommended to be in the lottery group.

panel gave on average higher IPL scores in 2015 than in the 2019 call. This makes it difficult and even unwise to compare scores of proposals of different disciplines, which were evaluated in distinct panels. These disciplinary differences are emphasised in the minimal consensus scores given to those proposals on the main list, *i.e.*, the proposals that were ultimately funded. The proposals from the main list evaluated by the Mathematics panel in 2015, for example, tend to have lower scores compared to the other panels (see S1 Table in the online supplement).

### Bayesian hierarchical model and ranking

Using the available budget from Table 1, the BR recommendations for the 2015 Social Sciences and Humanities panel can be found in Fig 3. Similar representations for all investigated panels in 2015 and 2019 can be found in S1 Figs in the online supplement. The expectations of the ranks (expected rank) are represented together with their 50% credible intervals. The provisional funding line (dashed blue line) is defined by allocating the available budget to the

**Table 2. Bayesian Ranking recommendations: number of proposals recommended to be part of the accepted or rejected proposals, or of the lottery group, for the selected panels (Life Sciences, Mathematics and Social Science/Humanities) and the two selected calls (2015 and 2019). N: Number.**

|  | N | Accepted (%) | Rejected (%) | Lottery (%) |
|---|---|---|---|---|
| **Call year: 2015** |  |  |  |  |
| Mathematics | 18 | 1 (5.6) | 17 (94.4) | 0 |
| Social Sciences/Humanities | 114 | 6 (5.3) | 100 (87.7) | 8 (7) |
| Life Sciences | 381 | 19 (5) | 341 (89.5) | 21 (5.5) |
| **Call year: 2019** |  |  |  |  |
| Mathematics | 16 | 1 (6.2) | 15 (93.8) | 0 |
| Social Sciences/Humanities | 122 | 5 (4.1) | 108 (88.5) | 9 (7.4) |
| Life Sciences | 355 | 22 (6.2) | 293 (82.5) | 40 (11.3) |

best ranked proposals until budget exhaustion is reached. The proposals with 50% credible interval crossing the provisional funding line are recommended to the lottery group. Table 2 summarises the recommendation of the Bayesian ranking for all panels of interest. For neither of the Mathematics panels a lottery group was recommended using a 50% credible interval: a clear jump from the best ranked proposal to the following proposals was observed. For the Social Sciences and Humanities panels, about 7% of the proposals ended up in the lottery group in both calls. For the Life Sciences, the call in 2015 resulted in a much smaller recommended lottery group as compared to the call in 2017.

**Research question I - comparison of consensus report and predictions.** Elements from the BHM can be used to predict the consensus report for proposal *i*: the overall mean scores $\bar{y}$ plus the posterior mean of $\theta_i$, *i.e.* the part of the proposal quality that is independent from the experts. Fig 4 shows a comparison of two quantities, the consensus reports and the BHM predictions. The proposals with a low consensus report tend to have a higher prediction while the opposite is true for the proposals with a high consensus report; the BHM predicted it to be lower. Hence, the spread in scores is larger after the consensus meeting than what would have been predicted from the individual evaluation reports. This is true, at least to some degree, for all panels and call years, but particularly pronounced for the Social Sciences and Humanities in 2019 and less for the Mathematics panel in 2019 and the Social Sciences and Humanities in 2015.

Some of the proposals in Fig 4 are outliers with very large discrepancies between prediction and actual consensus report (see the two highlighted Life Sciences proposals in the Figure). The proposal with the largest discrepancy in the Life Sciences panel in 2015 was predicted to have a consensus report of 79 using the BHM, but was given a consensus report of only 38. The individual evaluation reports this proposal received are summarised in Table 3 (case study one). Only the first expert gave this proposal lower than average scores. In the consensus meeting, this expert's negative evaluation outweighed the other more positive expert evaluations. In 2019, the largest discrepancy in the Life Sciences panel was found for a proposal that received a consensus report of 55 while the BHM predicted it to be 95. This proposal, case study two in Table 3, got high scores from all experts, so that the poor final consensus score cannot be explained from the data alone. Further investigations into these two special cases revealed that during the respective consensus meeting discussions it was remarked that specific supporting documents were missing which negatively affected the consensus scores and report. This additional information outlines the potential benefit of consensus meetings.

To further investigate the predictive capabilities of the individual evaluation reports, Fig 5 shows the consensus reports given to the proposals evaluated in 2015 together with the 95%

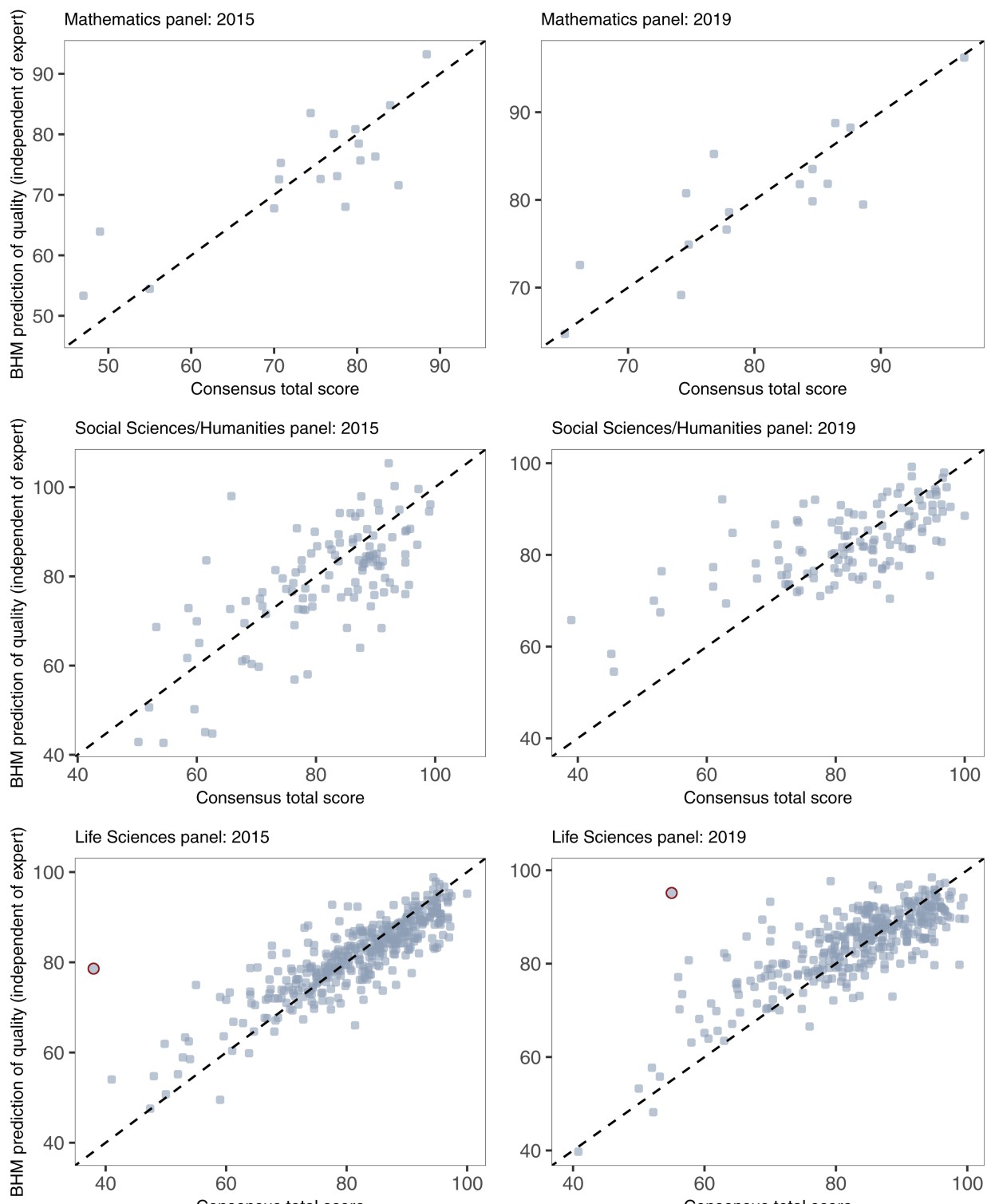

**Fig 4. Scatter plots comparing the consensus report and the prediction from the BHM for each panel.** Two outlier proposals in the Life Sciences panels, highlighted in dark red, are discussed more in detail in the text.

**Table 3. Individual evaluation reports and consensus report given to two specific outlier proposals, highlighted in Fig 4.** STE: scientific excellence, IPT: impact, IPL: implementation.

| Expert | STE score | IPT score | IPL score | Report |
|--------|-----------|-----------|-----------|--------|
| **Case study one (Life Sciences 2015)** | | | | |
| Expert 1 | 2.9 | 3.0 | 2.7 | 57.8 |
| Expert 2 | 4.7 | 4.4 | 4.6 | 91.8 |
| Expert 3 | 4.2 | 4.4 | 4.2 | 85.2 |
| Consensus | 2.0 | 2.0 | 1.5 | 38.0 |
| **Case study two (Life Sciences 2019)** | | | | |
| Expert 1 | 4.8 | 4.3 | 4.3 | 91.0 |
| Expert 2 | 4.5 | 4.5 | 3.5 | 86.0 |
| Expert 3 | 4.7 | 4.7 | 4.3 | 92.4 |
| Consensus | 2.7 | 3.0 | 2.5 | 55.0 |

credible intervals of the predictions from the BHM. The same Figure for the year 2019 can be found in the online supplement, S2 Fig. Whenever the consensus report falls within this range, *i.e.* the credible interval, it can be inferred that the consensus report aligns with what one would have anticipated based on the individual evaluation reports alone. Table 4 contains information on the share of proposals per panel and call that attained a consensus reports within, above, or below the predicted range. The smallest panel (Mathematics) does best, with the largest share of proposals with consensus reports in the expected range (around 70% in the "as expected" group), followed by the Life Sciences panels (≥ 50%). On the other hand, for the Social Sciences and Humanities panels only 27 to 39% of the consensus reports were as expected. In 2015, almost half of the proposals submitted to the Social Sciences and Humanities were corrected towards higher scores during the consensus meeting.

**Research question II - comparison of consensus final ranking and Bayesian ranking.** In Fig 6, the Bayesian ranking is compared to the official MSCA final ranking based on the consensus reports for all panels in 2015. The final ranking is shown against the 95% credible intervals of the ranks as modeled using the BHM. The color code informs on whether the official final ranking is within the 95% credible intervals of the Bayesian ranking. The same Figure for the calls in 2019 can be found in the online supplement, S2 Fig. Table 5 gives a summary on the share of proposals ranked within, better or worse than what would have been expected via the BR. The share of "as expected" for the ranking is generally higher than the share calculated for the consensus report as can be seen in Table 4 (apart from the Mathematics panel in 2015 which is rather small). While the MSCA panels provide a clear integer ranking, the BR is a continuous ranking. Therefore a perfect calibration of both rankings cannot be expected. The Bayesian ranking and the final ranking demonstrate the highest degree of similarity, *i.e,* overlap between final ranking and 95% credible interval, between the very best ranked and the very worst ranked proposals. However, for an important number of proposals within the large middle range, both rankings indicate a lack of alignment. Fig 7 represents the agreement of both rankings when splitting the proposals into various group sizes. For example, the first row in each sub-figure shows the agreement of the rankings for each percentile - do the BR and MSCA ranking agree on the 10% best proposals? do they agree on the group of proposals that were ranked among the 11 and 20% best? etc. The agreement is never consistently high (above 85% for the whole row). It also tends to be slightly higher for the best and worst ranked group, i.e., best quintile and fifth quintile, while the agreement in the middle groups is lower.

**Research question III - difference between BR recommendations and MSCA funding ranking.** Tables 6 and 7 show the agreement between the BR funding recommendation and

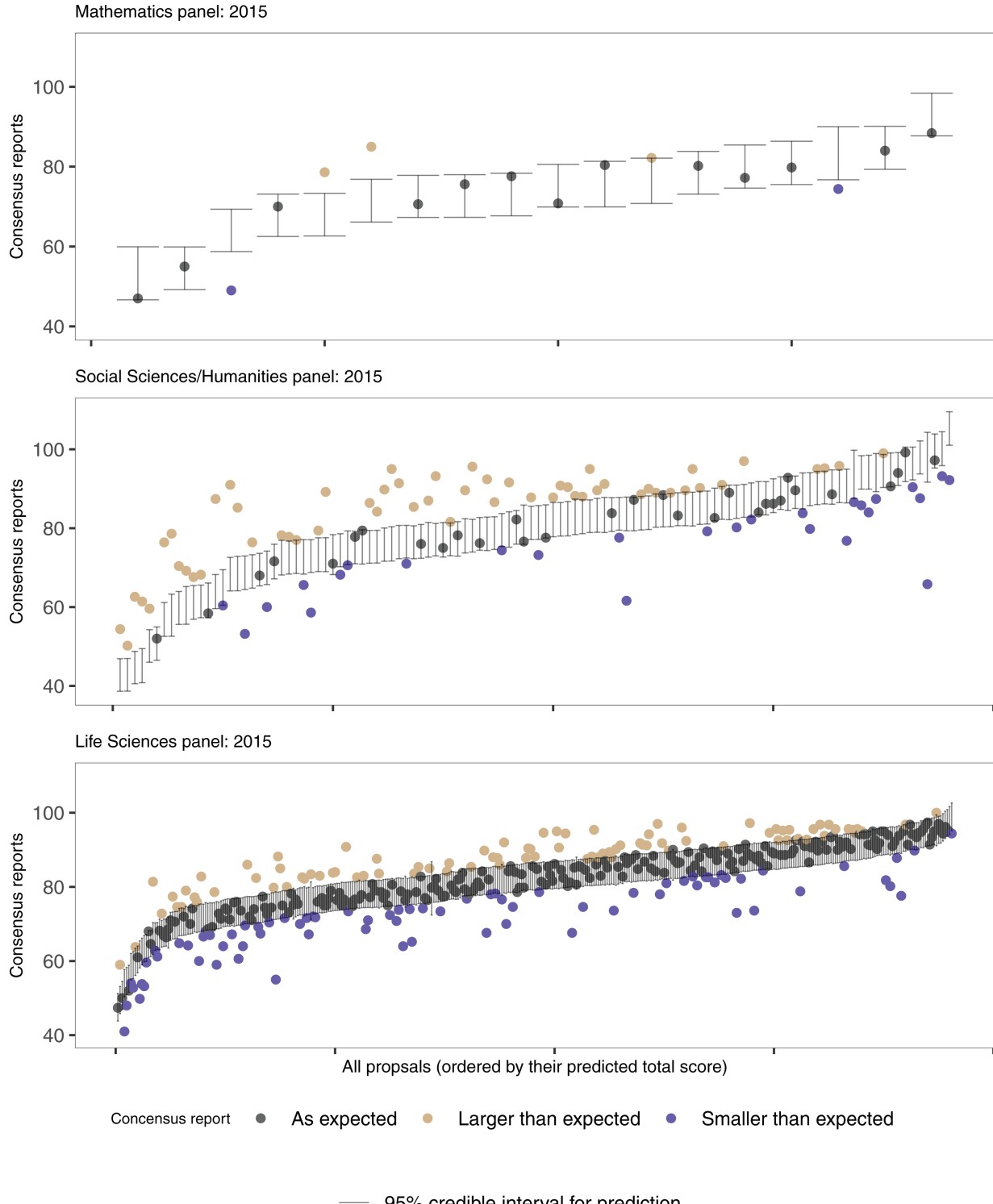

**Fig 5. 95% credible intervals of the predictions of the consensus report from the individual evaluation reports.** Note that even though the y-axes only start at 40, the scores could have been on a scale from 0 to 100. This further shows the skewness of the evaluation sores.

**Table 4. The share of proposals with consensus report within, above or below the 95% credible intervals of what would have been expected given the individual evaluation reports using the Bayesian hierarchical model.**

| | Consensus report | | |
|---|---|---|---|
| | As expected (%) | Larger than expected (%) | Smaller than expected (%) |
| **Call year: 2015** | | | |
| Mathematics | 13 (72.2) | 3 (16.7) | 2 (11.1) |
| Social Sciences/Humanities | 31 (27.2) | 56 (49.1) | 27 (23.7) |
| Life Sciences | 215 (56.4) | 88 (23.1) | 78 (20.5) |
| **Call year: 2019** | | | |
| Mathematics | 11 (68.8) | 2 (12.5) | 3 (18.8) |
| Social Sciences/Humanities | 47 (38.5) | 39 (32) | 36 (29.5) |
| Life Sciences | 179 (50.4) | 73 (20.6) | 103 (29) |

funding ranking after the MSCA consensus meetings. In the Mathematics panel only one proposal can be directly funded in 2015 and 2019. For both years, both ranking procedures find the same proposal to be funded. The BR does not recommend a lottery group, while the MSCA decides to put two proposals on the reserve list. By definition, the reserve list and the lottery group are inherently different. The result for the Mathematics panel suggests perfect agreement. This result is however driven by the low success rate and the small panel size. Looking at the Social Sciences and Humanities panel in 2015, among the 7 proposals on the MSCA main list, only one would have been accepted by the BR, two would be in the lottery group and four would have been rejected. Instead, the BR would have accepted five proposals that were rejected by the MSCA panel. Very similar trends are observed in 2019 and as for the Life Sciences panel. S2 Table in the online supplement investigates those proposals that were either rejected or accepted by the BR while the opposite was true in the final MSCA funding ranking in the Social Sciences and Humanities panel.

## Discussion

We set out to investigate the usability of an algorithmic approach as a substitution of the consensus meetings employed for the EU funding programmes. We used the individual evaluation reports given to proposals evaluated in the MSCA European Training Networks to predict the consensus reports, the final ranking as well as the funding ranking grouping the proposals into the main, the reserve and rejected list. Our extensive analysis concluded that predictions from the individual evaluation reports are poorly aligned with the consensus reports. Aggregating the individual evaluation reports into ranks, funding ranking groups or percentiles of ranks, seems to render the algorithmic approach more comparable. For example, for all investigated panels and call years, when using the BR to select the 10% best ranked proposals and comparing them to the 10% best ranked proposals in the MSCA final ranking, a substantial agreement of 84-95% is observed. For larger proportions of best ranked proposals, i.e., investigating the best 20% ranked *etc.*, agreement becomes worse. In our case study, the algorithmic approach aligned best for small panels with a low success rate.

Considering an alternate perspective, the results of our study made us wonder how useful the individual evaluation reports were. We would argue that, being such a poor predictor for the consensus report or funding ranking, they are not useful. A recent analysis of National Institutes of Health funding suggests, for example, that 47% of reviewers change their initial score after discussion (see S7 Table in [20]). Since we found higher agreement for the best and worst ranked proposals the individual total scores could help selecting the proposals that have low chances for acceptance. This was also suggested in previous research [21–23]. Therefore,

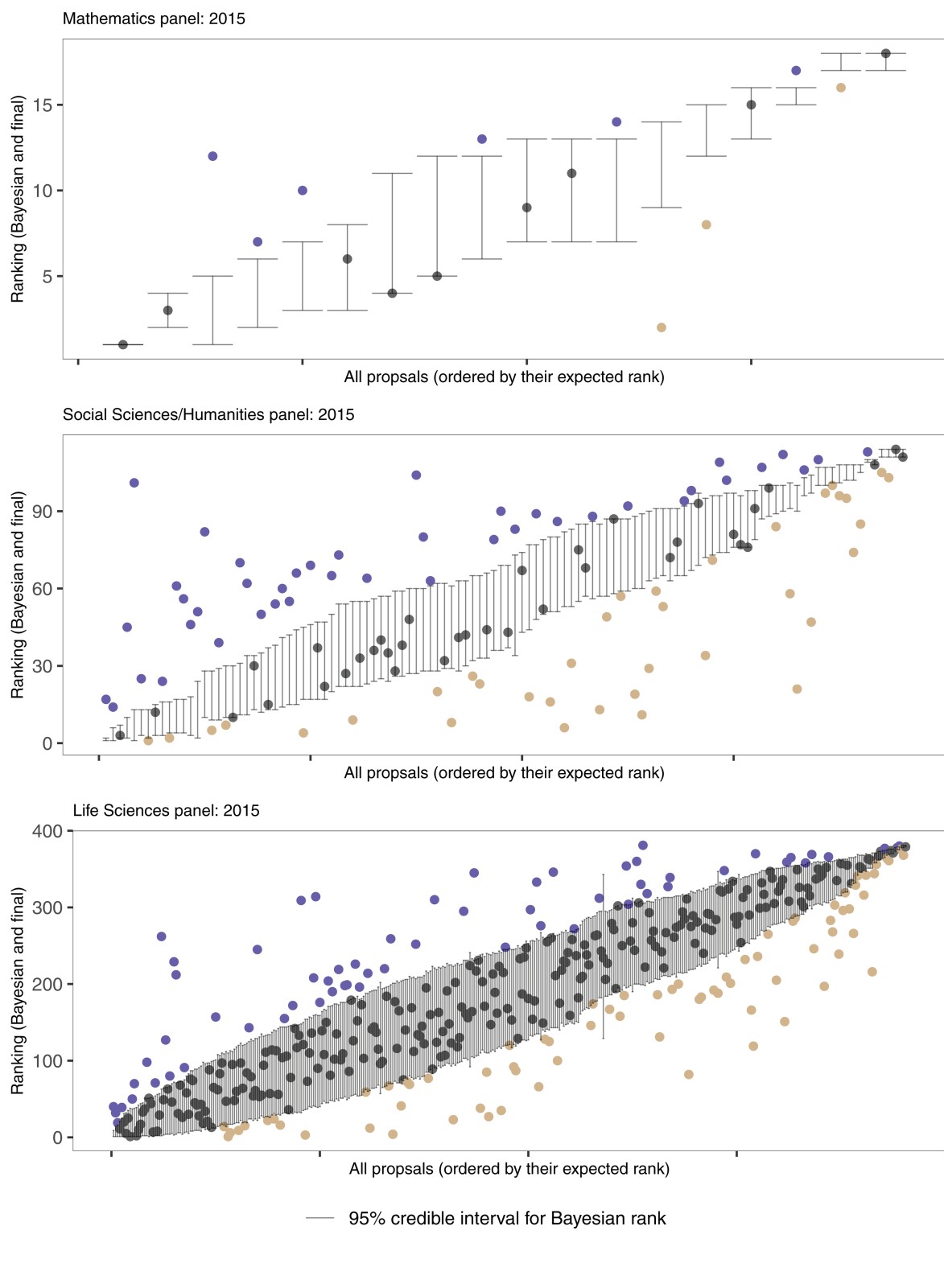

**Fig 6. The final and official ranking compared to the rank expected using the Bayesian hierarchical model with its 95% credible intervals for all three panels in 2015.** Note that even though the y-axes only start at 40, the scores could have been on a scale from 0 to 100. This further shows the skewness of the evaluation sores.

**Table 5. The share of proposals with official final ranking better or worse than what would have been expected given the individual evaluation reports used for the Bayesian ranking.**

|  | Final ranking | | |
|---|---|---|---|
|  | As expected (%) | Better than expected (%) | Worse than expected (%) |
| **Call year: 2015** | | | |
| Mathematics | 9 (50) | 3 (16.7) | 6 (33.3) |
| Social Sciences/Humanities | 36 (31.6) | 36 (31.6) | 42 (36.8) |
| Life Sciences | 245 (64.3) | 75 (19.7) | 61 (16) |
| **Call year: 2019** | | | |
| Mathematics | 13 (81.2) | 1 (6.2) | 2 (12.5) |
| Social Sciences/Humanities | 64 (52.5) | 27 (22.1) | 31 (25.4) |
| Life Sciences | 207 (58.3) | 78 (22) | 70 (19.7) |

individual evaluation reports and alike could potentially be useful to inform a triage of the worst proposals leading to a setting where only the remaining proposals, *i.e.*, a smaller subset of proposals, would be discussed in a panel meeting. As such, the effort of the experts and panel members would be used more efficiently. The exact proportion of triaged proposals needed to make the process cost-effective still has to be chosen and might depend on the evaluation process. Alternatively, the individual evaluation reports could inform the implementation of a modified lottery, as implemented for example by the Health Research Council of New Zealand for allocating their Explorer Grant funds [17]. Here, any proposal that was considered fundable after expert evaluation and based on a predefined criterion, would be subject to funding allocation by lottery.

Another interesting finding is that the variation in scores was larger after the consensus meeting compared to the spread that would have been anticipated from the individual evaluation reports alone. This suggests that although individual evaluation reports typically tend to be skewed towards higher scores, the discussion in the consensus meeting ensures that the entire (or at least a larger spectrum of the) grading scale was used. While this finding could support the case for consensus meetings in general, a case-by-case investigation of proposals with large discrepancies between the predicted and consensus reports could also further prompt questions about the dynamics within the consensus meeting itself [22]. The specific case studies we investigated here (in Table 3) would again support the consensus meeting, since it helped reveal the incompleteness of the submitted proposal documentation. However, we assume that integrating an algorithmic approach instead of a meeting in the decision making process could potentially eliminate a source of subjectivity, bias and uncertainty. Indeed, the Bayesian ranking combined with a lottery element was initially suggested to avoid lengthy and often biased panel discussions to discriminate those proposals that are neither clearly competitive nor clearly non competitive [19]. This assumption remains to be tested.

## Limitations

Our study is not without limitations. First of all, it is a purely exploratory analysis and no causal claims can be made. While we observe, in part, large discrepancies between the individual evaluation reports and the consensus reports, we cannot explain the latter with the data at our disposal. Our analysis focused on a very specific MSCA funding scheme, on two specific call years and tree specific scientific panels. The evaluation process of MSCA research proposals is based on three criteria (STE, IPT and IPL) scored on a scale from 0 to 5 which are

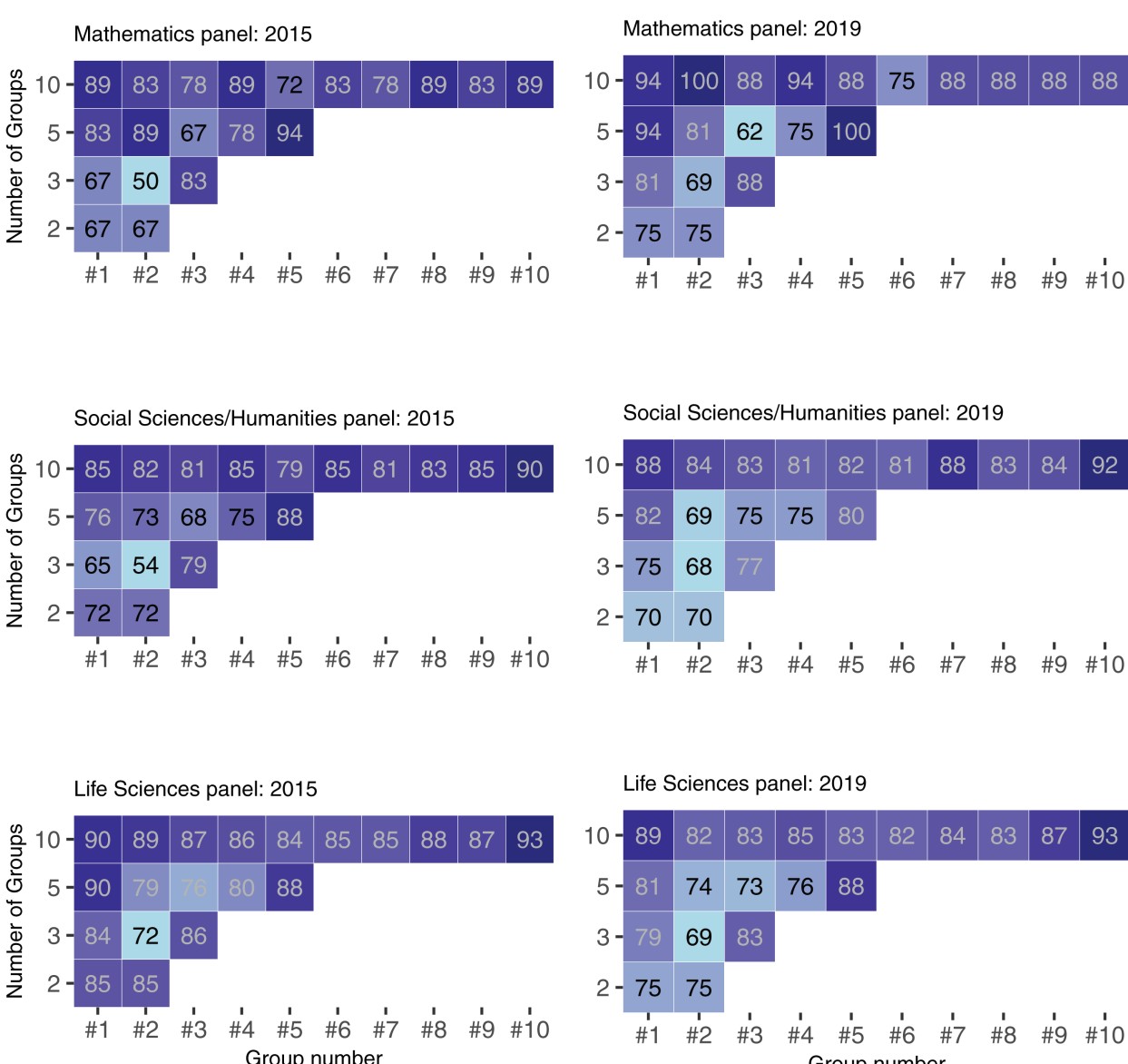

**Fig 7. Percentage of agreement between Bayesian Ranking and the official ranking for different group sizes.** For example, do the BR and official ranking agree on the 10% best ranked proposals? This is done for each panel in both call years.

summarised into a total score using the formula 50%STE + 30%IPT + 20%IPL converted into the individual evaluation report or the consensus report, on a scale from 0 to 100. The specificity of the included panels and calls and of the MSCA evaluation process makes the findings less generalisable, especially since differences in the process of weighting criteria scores into a total score could potentially penalise certain applicant groups [20]. A follow-up analysis on the newer Framework Programme, Horizon Europe, could help further investigate the generalisability of the findings. Another limitation of our approach is that the distribution of all scores was found to be skewed towards higher scores which makes it harder for any type of statistical model or algorithm to discriminate between proposals to fund or to reject.

**Table 6. Agreement for the panels in 2015, with the counts in the different funding ranking groups.**

| | BR recommendation | Main list | Reserve list | Rejected |
|---|---|---|---|---|
| Mathematics (88.9%) | | | | |
| | Accepted | 1 | 0 | 0 |
| | Lottery | 0 | 0 | 0 |
| | Rejected | 0 | 2 | 15 |
| Social Sciences/Humanities (82.5%) | | | | |
| | Accepted | 1 | 0 | 5 |
| | Lottery | 2 | 0 | 6 |
| | Rejected | 4 | 3 | 93 |
| Life Sciences (88.2%) | | | | |
| | Accepted | 8 | 3 | 8 |
| | Lottery | 2 | 4 | 15 |
| | Rejected | 14 | 3 | 324 |

**Table 7. Agreement for the panels in 2019, with the counts in the different funding ranking groups.**

| | BR recommendation | Main list | Reserve list | Rejected |
|---|---|---|---|---|
| Mathematics (93.8%) | | | | |
| | Accepted | 1 | 0 | 0 |
| | Lottery | 0 | 0 | 0 |
| | Rejected | 0 | 1 | 14 |
| Social Sciences/Humanities (86.1%) | | | | |
| | Accepted | 2 | 0 | 3 |
| | Lottery | 2 | 2 | 5 |
| | Rejected | 6 | 1 | 101 |
| Life Sciences (80.3%) | | | | |
| | Accepted | 7 | 3 | 12 |
| | Lottery | 10 | 2 | 28 |
| | Rejected | 10 | 7 | 276 |

## Conclusions

Our study offers novel insights into the dynamics of consensus meetings and the potential of algorithmic approaches to effectively aggregate individual expert evaluations as an alternative to traditional consensus meetings. In theory, the European commission could implement an algorithmic approach to summarise the evaluation reports instead of consensus meetings. It would surely make the process more efficient and less labour intensive for the experts. However, this would also put a lot more weight on the expert's individual evaluation, which are known to be, at least to a certain extend, unreliable, subjective and biases. Rather then abolishing the consensus meetings, it could be suggested to use the individual evaluation reports for a triage or similar prior to the consensus meeting. To decrease the uncertainty of the results extracted from the Bayesian hierarchical model one might suggest increasing the number of experts per proposal. However, as discussed by [24], a ridiculous number of experts would be needed to achieve small enough precision to make unambiguous funding decisions. Our recommendation for the commission and similar funding agencies would be to systematically scrutinize their funding evaluation and allocation system, to try and reduce the uncertainty of the process as much as possible while ensuring to not introduce any social disparities [25]. They should also experiment with different interventions on the funding call design or evaluation process to test whether the predictive capabilities of the individual scores improves. Additionally, longitudinal studies relating funding rankings to outcomes of funded projects could help test the underlying assumption that grant peer review effectively estimates the true quality of the proposals, which can only be judged retrospectively [26]. Some residual

uncertainty can however not be avoided nor eliminated. This is when elements such as a modified lottery [16], as also recommended by the Bayesian ranking methodology, should be considered. Further research and experimentation could help understand which changes should be applied to the funding and evaluation processes to ensure that the implementation of an algorithmic approach coupled with a lottery is accompanied with an increase in efficiency and a decrease in time spent preparing and evaluating grant proposals [27,28].

## Author contributions

**Conceptualization:** Rachel Heyard, David G. Pina, Ivan Buljan, Ana Marušić.

**Data curation:** David G. Pina.

**Methodology:** Rachel Heyard.

**Software:** Rachel Heyard.

**Visualization:** Rachel Heyard.

**Writing – original draft:** Rachel Heyard.

**Writing – review & editing:** Rachel Heyard, David G. Pina, Ivan Buljan, Ana Marušić.

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
