## [Decision Letter · Decision Letter 0]

30 Jul 2024

PONE-D-24-22791Black box ‘consensus meeting’: Describing the differences between MSCA consensus meetings and Bayesian ranking recommendationsPLOS ONE

Dear Dr. Heyard,

Thank you for submitting your manuscript to PLOS ONE. After careful consideration, we feel that it has merit but does not fully meet PLOS ONE’s publication criteria as it currently stands. Therefore, we invite you to submit a revised version of the manuscript that addresses the points raised during the review process.

We look forward to receiving your revised manuscript.

Kind regards,

Annesha Sil, Ph.D.

Associate Editor

PLOS ONE

[Rachel Heyard conceptualised and implemented the Bayesian ranking methodology used here while she was employed at the Swiss National Science Foundation. David G. Pina is employed by the European Research Executive Agency. Ivan Buljan has no competing interests to declare. Ana Marušić occasionally serves as an expert evaluator of grant proposals for the European Commission.]. 

Additional Editor Comments (if provided):

Reviewers' comments:

Reviewer's Responses to Questions

**Comments to the Author**

1. Is the manuscript technically sound, and do the data support the conclusions?

Reviewer #1: Yes

Reviewer #2: Yes

Reviewer #3: Partly

2. Has the statistical analysis been performed appropriately and rigorously? 

Reviewer #1: I Don't Know

Reviewer #2: Yes

Reviewer #3: Yes

3. Have the authors made all data underlying the findings in their manuscript fully available?

Reviewer #1: Yes

Reviewer #2: Yes

Reviewer #3: Yes

4. Is the manuscript presented in an intelligible fashion and written in standard English?

Reviewer #1: Yes

Reviewer #2: Yes

Reviewer #3: Yes

5. Review Comments to the Author

Reviewer #1: "The title of the paper seems a bit too jargon-laced, which might impact its visibility and downplay its importance. Perhaps a title that better captures the big picture upsides of this analysis, about algorithmic vs. grant panel approaches, and impact on science funding policy would be more fruitful."

Black Box ‘Consensus Meeting’: Describing the Differences Between MSCA Consensus Meetings and Bayesian Ranking Recommendations

This paper provides a cogent contribution to questions about the roles of consensus meetings in grant review, their relationship to individual reviewer reports, lotteries, and considers an alternative more ‘algorithmic’ approach. This study is interesting and carefully crafted, and I recommend it for publication. I have a number of suggested revisions, all of which are fairly minor. Thank you to the authors for their contribution to science funding policy methodology.

(1) There are a few references missing in the framing of the paper. Some work has been done on consensus meetings and reducing discrepancies caused by gender bias (Bol, T., de Vaan, M. and van de Rijt, A. [2022]: ‘Gender-Equal Funding Rates Conceal Unequal Evaluations’, Research Policy, 51, available at <doi.org 10.1016="" 2021.104399="" j.respol.="">.), and some work has been done on the inefficiencies of consensus meetings (Erosheva, E., Grant, S., Chen, M., Lindner, M., Nakamura, R. and Lee, C. J. [2020]: ‘NIH Peer Review: Criterion Scores Completely Account for Racial Disparities in Overall Impact Scores’, Science Advances, 6, available at <doi.org 10.1126="" sciadv.aaz4868="">.) and their implications for lotteries (Lee, C. J., Grant, S. and Erosheva, E. [2020]: ‘Alternative Grant Models Might Perpetuate Black–White Funding Gaps’, The Lancet, 396, pp. 955–56.).

(2) “for some an additional expert had to be consulted to find a consensus.”

- For clarification, would the additional expert be brought in only in cases with large discrepancies in initial scores?

(3) “to reach a consensus…”

- Is the purpose of these meetings to reach a consensus? If so, this seems worth noting (with specific reference to instructions to reviewers, if available/possible). Some grant review panels aren’t aiming at consensus explicitly, but rather just giving referees chances to hear their colleagues opinions.

(4) “with the most important budget…”

- Does this mean the largest budget?

(5) It might be interesting to briefly note that there don’t seem to be large differences of acceptance/reserve/reject rates despite changes in the amount of reviews required differing by quite a bit.

(6) “In this setting, the individual evaluation report yij can be interpreted as the expert’s best guess or estimation of the proposal’s true quality θi”

- I understand that this is a controversial point and don’t expect the authors to make major revisions to their methodology on this point, but I do think its worth highlighting that this conception of true quality is controversial. For example, see Shaw, J. 2023. “Peer Review, Innovation, and Predicting the Future of Science: The Scope of Lotteries in Science Funding Policy.” Philosophy of Science, 90(5): 1297-1306 for the argument that true quality is really a retrospective judgment, rather than expert’s best guess, that provides a proposals true quality (although I am adding my interpretation to this paper).

(7) “the budget generally determines the number of proposals x that can be funded”

- I assume that the budget is fixed prior to the reviewing process, so there’s no wiggle room for budget negotiations? If so, this could be made explicit.

(8) “Each scientific panel has its own grading behaviour”

- Maybe a word or two here adding some specifics would be helpful here.

(9) “We would argue that, being such a poor predictor for the consensus report or funding ranking, they are not useful”

- It might be worth noting related studies. In the Erosheva et al. study mentioned above, they found that 43% of reviewers changed their preliminary scores as a result of panel discussions.

(10) “None of our findings can be generalised to other funding agencies or funding schemes as the analysis focused on a very specific MSCA funding scheme, on two specific call years and tree specific scientific panels”

- This might be overstating the case. Perhaps a more modest warning, like saying that generalizations should be cautious to consider the specifics of the MSCA funding scheme, would be appropriate.</doi.org></doi.org>

Reviewer #2: The manuscript highlights an important and underdeveloped area of funding practices. The authors have adequately reported the uncertainty in this field and suitably reference past evidence in this space. I have some additional comments under each of the manuscript headings, see below.

For those who may not know what the MSCCA is I think this needs to be reported in full in the title.

Abstract: There is no background in the abstract to put this research into context. Understanding the unmet need and the reason for the research/study would strengthen the abstract.

Introduction: Providing some context around describing the allocation of funding as simplistic and ignoring the underlying uncertainty needs elaboration. It is not clear what this is referring to and given the existing evidence in this space, I would suggest updating the 2008 reference used. The authors adequately provide context around alternative approaches and how this current work fits into the wider context. Perhaps, providing the three research questions to the introduction (or methods) would make it easier to see the purpose of the analysis.

Methods: the authors provide a thorough account of the approach taken. Who conducted the analysis and the software used for analysis is not reported though, which would help others to replicate in the future. Perhaps, move the research questions to the beginning of the methods section so the reader is able to see this first. Although there is a data analysis subheading there is minimal information provided, albeit a link to OSF protocol. Listing the questions outlined in the protocol would be very useful in the manuscript (at the end of the introduction or as part of the methods). Having them earlier in the manuscript would then support the information detailed in Box 1.

Why was the data from 2015-2019? As this is now 5 years ago and have processes/practices changed during this period. Explaining why the analysis was with 5 year old data needs to be reported within the manuscript to be able to place the analysis in the wider context.

The three research questions are adequately reported in the results section, along with tables and figures to further explain the results of the analysis.

The discussion relates back to the three research questions, although I'm unsure of the second paragraph and the justifications around the usefulness of individual evaluation reports. Further explanation is required, to back up the statement based on the results of the analysis. It is also challenging to understand what this paragraph is trying to say, and its relation to existing evidence. The authors provide limitations of the analysis, although the last three sentences appear out of place. It would be better placed as a final paragraph to summarise the analysis in the wider context of funding allocation, how funders can use this in future considerations (along with caveats) and what does it mean in terms of bias. As noted, points raised in the introduction around the challenges with

funding allocation (inefficient, biased, 'black box') need to be picked up in the discussion. In particular, the 'black box' terminology, mentioned in the introduction is not mentioned throughout the manuscript.

Using algorithms to attempt to remove bias and inefficiencies also has limitations, which is mentioned in the existing literature. Balancing funding allocation approaches is challenging, yet can complement each other given the right context. I feel the authors could pick up these opportunities in the manuscript more and tense out how and in what circumstances it would be acceptable to use algorithms to inform funding allocation processes and ensure these practices are transparent, fair and of high quality. How this analysis aligns and offers these opportunities more widely could be strengthened in the manuscript.

Minor corrections: There are typo errors on the figures such as 1,006 instead of 1'006 and 7,870 instead of 7'870

Typo error - [Pina et al., 2015, Graves et al. [2011], Erosheva et al. [2021]].

Reviewer #3: Technical Soundness:

The manuscript is partly technically sound. The methodology, involving a Bayesian hierarchical model to compare individual evaluation reports (IER) and consensus meeting reports (CR), is robust and well-chosen for the research questions posed. The study design includes appropriate controls and comparisons, with an adequate sample size across multiple panels and call years. However, while the Bayesian model is rigorously applied, the generalizability of the findings to other funding schemes and contexts remains uncertain. Detailed case studies and further validation of the model using additional datasets could enhance the robustness of the conclusions.

Statistical Analysis:

The statistical analysis has been performed appropriately and rigorously. The use of a Bayesian hierarchical model to account for variability and uncertainty in the data is commendable. The manuscript provides a clear explanation of the statistical methods used, and the comparative analysis between the Bayesian ranking and consensus meetings is thorough and well-supported by the data.

Data Availability:

The authors have made all data underlying the findings in their manuscript fully available. This transparency is crucial for the reproducibility of the research. The data is openly accessible through the provided links, ensuring that other researchers can verify and build upon the study's findings.

Manuscript Presentation:

The manuscript is presented in an intelligible fashion and written in standard English. The structure is logical, and the writing is clear, making the complex statistical methodologies and findings accessible to a broad audience. However, the manuscript could benefit from a more detailed discussion on the potential biases in both the Bayesian model and consensus meetings.

Additional Comments:

Generalizability: The study focuses on the MSCA funding scheme and a limited number of panels. A discussion on how the findings could be generalized to other funding mechanisms would be beneficial.

Bias Analysis: A thorough analysis of potential biases in expert evaluations and consensus discussions could provide valuable insights and strengthen the manuscript's conclusions.

Future Research: Outlining specific future research directions to test the applicability of the findings in other contexts and with different types of funding mechanisms would be a valuable addition.

While the manuscript is technically sound and well-written, addressing a few areas could enhance its clarity and robustness. Specific improvements include a more detailed discussion on the generalizability of findings, potential biases in both the Bayesian model and consensus meetings, and future research directions. Addressing these points would make the manuscript even stronger.

6. PLOS authors have the option to publish the peer review history of their article (what does this mean?). If published, this will include your full peer review and any attached files.

Reviewer #1: No

Reviewer #2: No

Reviewer #3: No

---

## [Author Response · Author response to Decision Letter 1]

6 Sep 2024

The detailed response to the reviewer comments was uploaded as PDF.

---

## [Decision Letter · Decision Letter 1]

15 Oct 2024

PONE-D-24-22791R1Assessing the potential of a Bayesian ranking as an alternative to consensus meetings for decision making in research funding: A Case Study of Marie Skłodowska-Curie ActionsPLOS ONE

Dear Dr. Heyard,

Thank you for submitting your manuscript to PLOS ONE. After careful consideration, we feel that it has merit but does not fully meet PLOS ONE’s publication criteria as it currently stands. Therefore, we invite you to submit a revised version of the manuscript that addresses the points raised during the review process.

We look forward to receiving your revised manuscript.

Kind regards,

Irfan Ullah, PhD

Academic Editor

PLOS ONE

Journal Requirements:

Reviewers' comments:

Reviewer's Responses to Questions

**Comments to the Author**

1. If the authors have adequately addressed your comments raised in a previous round of review and you feel that this manuscript is now acceptable for publication, you may indicate that here to bypass the “Comments to the Author” section, enter your conflict of interest statement in the “Confidential to Editor” section, and submit your "Accept" recommendation.

Reviewer #2: All comments have been addressed

Reviewer #4: (No Response)

2. Is the manuscript technically sound, and do the data support the conclusions?

Reviewer #2: Yes

Reviewer #4: Yes

3. Has the statistical analysis been performed appropriately and rigorously? 

Reviewer #2: N/A

Reviewer #4: Yes

4. Have the authors made all data underlying the findings in their manuscript fully available?

Reviewer #2: Yes

Reviewer #4: Yes

5. Is the manuscript presented in an intelligible fashion and written in standard English?

Reviewer #2: Yes

Reviewer #4: Yes

6. Review Comments to the Author

Reviewer #2: (No Response)

Reviewer #4: 1 In the introduction section, the author is requested to discuss the theoretical and methodological foundations of existing research, the gaps in existing research, and the areas in which this article will advance existing theories and methods.

2 In the conclusion section, please have the author discuss the theoretical and methodological contributions of this article.

7. PLOS authors have the option to publish the peer review history of their article (what does this mean?). If published, this will include your full peer review and any attached files.

Reviewer #2: **Yes: **A J Blatch-Jones

Reviewer #4: No

---

## [Author Response · Author response to Decision Letter 2]

5 Nov 2024

Thank you for reviewing our paper. We hereby submit a revised version of our manuscript. Please find below a point by point response to the comments by the reviewer. We also provide a version of our manuscript highlighting the changes we made.

Reviewer

In the introduction section, the author is requested to discuss the theoretical and methodological foundations of existing research, the gaps in existing research, and the areas in which this article will advance existing theories and methods.

Reply: Thank you for your comment. We believe our introduction discussed existing research and the motivations for our study. The gaps in the existing research are now emphasised with the sentence we added at the end of the second paragraph. The specific areas that our study will advance are explained at the end of the third paragraph:

We specifically use the BR to determine the contexts in which such a substitution would be advantageous and the scenarios in which it may be less suitable. This study provides valuable insights into whether the outcomes or decisions derived from the consensus meeting can be predicted from the individual evaluation scores alone, or whether the dynamics of the consensus meetings are unpredictable solely from the individual evaluation scores. Our results contribute to enhancing our knowledge of the feasibility and effectiveness of such algorithmic approaches in grant proposal evaluations, ultimately facilitating evidence-based decision-making in research funding allocation and making the process more transparent and efficient for expert evaluators, panel members and applicants.

--------

In the conclusion section, please have the author discuss the theoretical and methodological contributions of this article

Reply: We believe that the contributions of our article are described in the Discussion, but we now added an opening sentence to the Conclusions that further emphasises the contribution: Our study offers novel insights into the dynamics of consensus meetings and the potential of algorithmic approaches to effectively aggregate individual expert evaluations as an alternative to traditional consensus meetings. We also did minor changes to the Conclusions section to improve the flow.

---

## [Decision Letter · Decision Letter 2]

5 Jan 2025

Assessing the potential of a Bayesian ranking as an alternative to consensus meetings for decision making in research funding: A Case Study of Marie Skłodowska-Curie Actions

PONE-D-24-22791R2

Dear Dr. Heyard,

We’re pleased to inform you that your manuscript has been judged scientifically suitable for publication and will be formally accepted for publication once it meets all outstanding technical requirements.

Kind regards,

Sergio A. Useche, Ph.D.

Academic Editor

PLOS ONE

Additional Editor Comments (optional):

The paper can be accepted in its current form. Thanks for the amendments and clarifications.

Reviewers' comments:

Reviewer's Responses to Questions

**Comments to the Author**

1. If the authors have adequately addressed your comments raised in a previous round of review and you feel that this manuscript is now acceptable for publication, you may indicate that here to bypass the “Comments to the Author” section, enter your conflict of interest statement in the “Confidential to Editor” section, and submit your "Accept" recommendation.

Reviewer #2: All comments have been addressed

2. Is the manuscript technically sound, and do the data support the conclusions?

Reviewer #2: Yes

3. Has the statistical analysis been performed appropriately and rigorously? 

Reviewer #2: I Don't Know

4. Have the authors made all data underlying the findings in their manuscript fully available?

Reviewer #2: Yes

5. Is the manuscript presented in an intelligible fashion and written in standard English?

Reviewer #2: Yes

6. Review Comments to the Author

Reviewer #2: Thank you for addressing the feedback, I have no further comments to add to the revised version. The authors have revised the manuscript according to the feedback they have received.

7. PLOS authors have the option to publish the peer review history of their article (what does this mean?). If published, this will include your full peer review and any attached files.

Reviewer #2: **Yes: **Amanda Blatch-Jones

---

## [Editor Report · Acceptance letter]

PONE-D-24-22791R2

PLOS ONE

Dear Dr. Heyard,

I'm pleased to inform you that your manuscript has been deemed suitable for publication in PLOS ONE. Congratulations! Your manuscript is now being handed over to our production team.

Kind regards,

on behalf of

Dr. Sergio A. Useche

Academic Editor

PLOS ONE